# Probiotic Supplementation and Human Milk Cytokine Profiles in Japanese Women: A Retrospective Study from an Open-Label Pilot Study

**DOI:** 10.3390/nu13072285

**Published:** 2021-06-30

**Authors:** Tomoki Takahashi, Hirofumi Fukudome, Hiroshi M. Ueno, Shiomi Watanabe-Matsuhashi, Taku Nakano, Toshiya Kobayashi, Kayoko Ishimaru, Atsuhito Nakao

**Affiliations:** 1Research and Development Department, Bean Stalk Snow Co., Ltd., Saitama 350-1165, Japan; h-fuku@meg-snow.com (H.F.); hiroshi-ueno@meg-snow.com (H.M.U.); s-matsuhashi@meg-snow.com (S.W.-M.); nakano@meg-snow.com (T.N.); t-kobayashi@meg-snow.com (T.K.); 2Department of Immunology, Faculty of Medicine, University of Yamanashi, Yamanashi 409-3898, Japan; ikayoko@yamanashi.ac.jp (K.I.); anakao@yamanashi.ac.jp (A.N.)

**Keywords:** human milk, cytokine, probiotics, IL-10, *Lactobacillus casei*, *Bifidobacterium longum*, *Bacillus coagulans*, multiplex assays, pilot study, lactating mothers

## Abstract

The benefits of probiotic supplementation to lactating mothers on human milk cytokines are inconclusive. Thus, we performed a comprehensive open-label pilot trial analysis of 27 human milk cytokines in lactating women with allergies (one to three months postpartum) to determine the effect of supplementation with a mixture of new probiotic strains. Participants voluntarily joined the probiotic (*n* = 41) or no supplementation control (*n* = 19) groups. The probiotic group took three probiotic tablets (*Lactobacillus casei* LC5, *Bifidobacterium longum* BG7, and *Bacillus coagulans* SANK70258) daily for one to three months postpartum. Milk samples were collected at one, two, and three months postpartum, and cytokine levels were measured using multiplex assays. The effects were analyzed using multivariate regression models. Eleven cytokines showed a positive rate of over 50% in the milk samples throughout testing in both groups. The positive rates of IL-1 receptor antagonist and IL-7 changed significantly with lactation progression in logistic regression models after adjusting for time and supplementation, whereas rates of other cytokines showed no significant differences. The lactational change patterns of IL-10 concentrations differed significantly between the two groups. A short-term supplementation of probiotics affects human milk cytokine levels in lactating women with a possible placebo effect still existing. Future placebo-controlled studies are needed to support these results, based on the estimated sample sizes in this study.

## 1. Introduction

Breastfeeding provides the ideal nutrition for infant growth and development. It can protect against gastrointestinal or respiratory tract infections [1] and reduce the risk of diarrhea morbidity or mortality in early infancy [2]. Human milk contains various bioactive proteins related to the immune system, such as cytokines, chemokines, and growth factors; therefore, it has been studied as one of the key environmental factors influencing immunological development in early infancy. Nevertheless, there is conflicting evidence for variation in human milk cytokines throughout breastfeeding and in relation to infant development. Many factors may partially introduce variations in the composition of human milk since a large variety of individual components possess immunological activity [3,4]. There are hundreds of known immunologically bioactive proteins, prebiotic oligosaccharides, membrane-associated lipid molecules, and microbiomes in existence [5]. Milk cytokines function as signaling molecules that promote and alleviate inflammatory responses and help stimulate the development of the immune system in newborns. Milk chemokines are possibly involved in maternal allergy and peripartum viral transmission, based on the results of research on milk secreted by HIV-infected mothers [6], women with mastitis [7], and allergies [8,9]. Growth factors as a part of infant nutrition may play a role in developing gastrointestinal tracts.

Dietary and probiotic supplementation to pregnant or lactating mothers has been studied as a promising option for altering the composition of immune components in breast milk and preventing the development of allergies in infants and children. Many trials of probiotic supplementation using single-entity products of a specific strain or mixtures have been conducted to determine the prevention of allergy development; cumulative meta-analytic evidence has suggested some protection against eczema [10]. TGF-β1 and immunoglobulin A (IgA) levels are high in the milk of mothers receiving *Bifidobacterium lactis* HN019 probiotics, and milk IgA levels are high in the milk of mothers receiving *Lactobacillus rhamnosus* HN001 [11]. As the beneficial effects of probiotics are inconclusive and appear to be strain-specific, each single strain or combination of strains should be individually investigated to determine their effect on immunological outcomes. When evaluating the immunological effects of supplementation with a new probiotic strain or combination of strains, the pilot study is important for improving the quality and efficiency of the main study [12]. Our previous pilot study suggested that a mixture of new probiotic strains (*Lactobacillus casei* LC5, *Bifidobacterium longum* BG7, and *Bacillus coagulans* SANK70258) was tolerated in lactating mothers regarding no dropout and 91.5% adherence; further, it had possibly affected TGF-β and IgA levels in milk [13]. Previous studies have investigated only a limited range of cytokines [11,14,15,16,17,18]; considering the complex network interaction among cytokines, investigating a greater number of cytokines may help explain the contradiction in results between these studies.

In this study, 27 cytokines were comprehensively measured in breast milk samples collected in a previous study to calculate the sample size necessary for a full analysis and determine the effect of the new probiotic strains, *L. casei* LC5, *B. longum* BG7, and *B. coagulans* SANK70258, on other breast milk cytokines.

## 2. Materials and Methods

### 2.1. Participants and Study Design

The open-label pilot study has been described in detail previously [13]. Briefly, we recruited 63 lactating women from the Tokyo metropolitan area and informed them of the content and intention of the study. In this open-label pilot trial conducted between April 2013 and December 2013, 60 healthy lactating women volunteered to participate at the Gonohashi Obstetrics and Gynecology Hospital (Tokyo, Japan). Three additional women dropped out of the study because of a lack of informed consent. To lessen the burden and dropout rate, the participants self-selected to join either the probiotic group (*n* = 41) or the no supplementation control group (*n* = 19). The probiotic group received three probiotics tablets daily after breakfast containing 5 × 10^9^ CFU of *L. casei* LC5 (CJ Japan Co., Ltd., Tokyo, Japan), 5 × 10^9^ CFU of *B. longum* BG7 (CJ Japan Co., Ltd., Tokyo, Japan), and 2 × 10^8^ CFU of *B. coagulans* SANK70258 (Mitsubishi-kagaku Foods Co., Ltd., Tokyo, Japan), respectively—for about two months starting from one to three months postpartum. Detailed information regarding the probiotic tablets is shown in Table 1. Three combinations of probiotics were selected for this study [13]. *L. casei* LC5 is isolated from cheese [19]. A mixture containing LC5 reduces lipopolysaccharide-induced NO production from the murine RAW264.7 cells, indicating that LC5 involves in anti-inflammatory activity [20]. *B. longum* BG7 is isolated from infant feces [21]. BG7 suppresses IL-8 production in human gastric cell lines with *Helicobacter pylori* [22], implying that BG7 can modulate cytokine production in bacterial infection. *B. coagulans* SANK70258 is isolated from green molt [23]. SANK70258 improves the intestinal environment in healthy adults, as evidenced by changes in bifidobacteria and clostridia [23]. Given this information, it is reasonable to assume these probiotic strains may exert beneficial effects on health by modifying the immune system. Furthermore, a previous report indicates that supplementation with a mixture of different probiotic genera, namely *Bifidobacterium* and *Lactobacillus*, is promising in preventing allergy in infants; in this report, the subjects were supplemented with ≥ 10^9^ CFU in many trials [24]. In addition, to increase adherence by providing a perceptible benefit to the mother herself, we included 10^8^ CFU tablets of *B. coagulans* SANK70258, this being shown to help improve bowel movements [23]. Thus, we studied a combination of these three species with 5 × 10^9^ CFU of *L. casei* LC5, 5 × 10^9^ CFU of *B. longum* BG7, and 2 × 10^8^ CFU of *B. coagulans* SANK70258. All participants maintained a diary record of the daily ingestion of the test tablets (only for the probiotic group), fermented foods, and medicines throughout the study.

### 2.2. Collection of Human Milk Samples

Human milk samples were collected manually at one, two, and three months postpartum by mothers after breastfeeding, using a breast pump (Yanase Waichi, Japan). Individual samples were collected in plastic tubes and stored at −80 °C for further analyses. The milk samples were thawed immediately before testing and were first centrifuged at 500× *g* for 5 min and then at 10,000× *g* for 10 min. Skimmed milk was extracted using pipettes and was subjected to further analysis.

### 2.3. Measurements of Cytokines and Chemokines in Human Milk

The Bio-Plex Pro™ Human Cytokine 27-plex Assay (Bio-Rad Laboratories, Inc., Tokyo, Japan) containing magnetic beads coated with antibodies against 27 cytokines and chemokines in a suspension array system, was used to measure the human milk cytokine levels. Interleukin (IL)-1β, IL-1 receptor antagonist (RA), IL-2, IL-4, IL-5, IL-6, IL-7, IL-8, IL-9, IL-10, IL-12 (p70), IL-13, IL-15, IL-17, eotaxin, fibroblast growth factor-basic (FGF-basic), granulocyte colony-stimulating factor (G-CSF), granulocyte-macrophage colony-stimulating factor (GM-CSF), interferon-gamma (IFN-γ), interferon-gamma induced protein 10 (IP-10), monocyte chemotactic protein-1 (MCP-1), macrophage inflammatory protein (MIP)-1α, MIP-1β, platelet-derived growth factor-BB (PDGF-BB), regulated on activation, normal T cell expressed and secreted (RANTES), tumor necrosis factor-α (TNF-α), and vascular endothelial growth factor (VEGF) levels were analyzed in milk samples collected at 1, 2, and 3 months postpartum. The components measured in this study are summarized in Appendix A. All skimmed milk samples were diluted 4-fold with sample diluent (50 µL skimmed milk + 150 µL sample diluent) before use. The assay was conducted according to the manufacturer’ s instructions. Briefly, 50 μL of the magnetic beads were vortexed for 30 s and then added to each well of the assay plates. The plate was washed with 100 μL wash buffer per well. The diluted skimmed milk samples, cytokine standards, and blank were vortexed for 5 s and were added to the well in duplicates of the plate. The assay plate was covered with sealing tape and incubated on a shaker at 850 ± 50 rpm for 30 min at 25 °C. After incubating, the plate was washed three times with 100 µL wash buffer. The detection antibodies were vortexed for 5 s and 25 µL was added to each well. The plate was covered and incubated on a shaker at 850 ± 50 rpm for 30 min at 25 °C. After detection antibody incubation, the plate was washed three times with 100 µL wash buffer. The streptavidin-phycoerythrin (SA-PE) solution was vortexed for 5 s and 50 µL was added to each well. After the SA-PE incubation, the plate was washed three times with 100 µL wash buffer. To resuspend beads for plate reading, 125 µL assay buffer was added to each well. The plate was covered with sealing tape and shakend at –25 °C, 850 ± 50 rpm for 30 s. After shaking, the assay plates were read using a Bio-Plex 200 System plate reader (Bio-Rad Laboratories) and determined using the Bio-Plex Manager software, version 5.0.

### 2.4. Data Analyses

The sample size was set at 60 subjects (30 per group), with 10 participants per predictor and 6 possible predictors allowance estimated for this study [25].

Baseline characteristics data were analyzed using the Mann-Whitney U test. Clinical history was analyzed using Pearson’s χ^2^ test. Positive rates of cytokines in the milk samples were analyzed as nominal data and concentrations were handled as continuous data. Positive rates in different groups at the same time points were analyzed by Fisher’s exact test, and those in the same group at different time points were analyzed using McNemar’s test. Cytokine concentrations in different groups at the same time points were analyzed by Mann-Whitney U test and those in the same groups at different time points were analyzed by Wilcoxon signed-rank sum test (Appendix A). Cytokines with a positive rate of less than 50% were considered as low quantitative values and were not included in subsequent analyses.

Differences in the effects of the supplements over time were analyzed using logistic regression and repeated measures ANOVA for positive rates and concentrations, respectively. One between-subjects factor (supplement: control and probiotic), one within-subjects factor (time: 1, 2 and 3 months postpartum), and their interaction were considered in these analyses. To validate the analysis, a Greenhouse-Geisser correction was used in the repeated ANOVA measurements because the Mauchly’s sphericity test used was inappropriate. All statistical analyses were performed using SPSS Statistics 26 (IBM Inc., Chicago, IL, USA). For the data visualization, generation of box plots, network analysis, and hierarchical heat maps, R software version 3. 6. 1 (R Foundation for Statistical Computing, Vienna, Austria) and the R package igraph were used, respectively. The levels of 24 cytokines were converted to common logarithm values to visualize the distribution of the data. The network analysis and hierarchical heat maps were derived from Pearson’s correlation coefficient. Unless otherwise noted, *p <* 0.05 was considered statistically significant.

### 2.5. Ethics

This study was conducted under the recommendations of the Ethical Guidelines for Clinical Research (Ministry of Health, Labor, and Welfare, Japan), and was approved by the Ethics Committee of the Faculty of Medicine, University of Yamanashi, (Yamanashi, Japan; receipt No. 1042, 2013). All subjects provided written informed consent in accordance with the Declaration of Helsinki. This study was registered with the University Hospital Medical Information Network Clinical Trials Registry (Available online: www.umin.ac.jp/ctr/, UMIN ID 000036059) (accessed on 1 March 2019).

## 3. Results

### 3.1. Background Characteristics of Mothers and Infants

The baseline values and maternal allergic disease status at one month postpartum are reported in Table 2. These indices were similar and did not differ significantly between the probiotic and no supplementation control groups.

### 3.2. Positive Rates of Cytokines in Human Milk

Standard curves fitted with 5-parameter logistic regression were successfully generated to calculate the positive rates and concentrations for 24 of the 27 cytokines measured except for IFN-γ, IL-15, and MCP-1. Regarding the positive rates, 11 of 24 cytokines were detected at a concentration of above 50% in the milk samples at all time points in both groups (*n* = 180). The positive rates did not differ between the control and probiotic groups at the same time points (Appendix A). In the probiotic group, IL-1RA, IL-6, and RANTES levels decreased significantly from one to two months and MIP-1α increased significantly from two to three months. IL-1RA levels significantly decreased from one to three months in both groups (Appendix A). Table 3 summarizes the effects of probiotic supplementation on the positive rates of cytokines. We analyzed the positive rates of cytokine levels as a function of time and intervention to clarify the effects of confounders using logistic regression. The positive rates of IL-1RA and IL-7 changed significantly over time (*p =* 0.0002 for IL-1RA and *p =* 0.026 for IL-7), whereas those of the cytokines did not change significantly with time and intervention.

### 3.3. Levels of Cytokines in Human Milk

We performed individual comparisons of cytokine concentrations as a crude analysis. IL-10 and IL-12 (p70) levels were significantly higher in the probiotic group than in the control group at one and two months (Appendix A and Appendix A). VEGF levels were significantly higher in the probiotic group than in the control group at one month. In the probiotic group, IP-10 levels decreased significantly from one to two and three months (Appendix A and Appendix A). However, IL-12 (p70) and VEGF levels increased significantly from one to two months and IL-8 levels increased significantly from two to three months. In the control group, IL-10 levels significantly increased from one to three months. IL-1RA levels showed a significant decrease from one to two and three months in both groups. Table 4 summarizes the effects of probiotic supplementation on the cytokine and chemokine levels, based on individual comparisons. To clarify the effects of confounders, we analyzed the cytokine levels as a function of time and intervention with repeated measures ANOVA. IL-10 levels were significantly affected by the intervention × time interaction (*p =* 0.015). The levels of IL-10, IL-12 (p70), and VEGF were significantly affected by the intervention after adjusting for time (*p =* 0.003 for IL-10, *p =* 0.003 for IL-12 (p70), and *p =* 0.041 for VEGF). IL-10 and VEGF levels also changed significantly over time (*p =* 0.001 for IL-10 and *p =* 0.011 for VEGF). However, the levels of the other cytokines did not change significantly with time and intervention. The topology of cytokine networks and their hierarchical heat maps were visually different between groups and lactating period (Appendix A).

## 4. Discussion

In this study, we analyzed the effects of probiotic supplementation on 27 cytokines in human milk. We found a different pattern of IL-10 levels at one to three months postpartum upon probiotic supplementation. Probiotic supplementation to pregnant and lactating mothers has been reported to alter the immune composition of breast milk [11,14,15,16,17,18]. In these studies, mothers were given probiotics from pregnancy to lactation, and the colostrum (0–7 days) and mature milk (one month or later) were collected to examine the relationship between the changes in the levels of breast milk cytokines such as IgA, TGF-β, and soluble CD14, using individual assays.

We used a multiplex assay to measure 27 cytokines, chemokines, and growth factors, which is higher than the number of analytes reported in previous studies. Cytokines co-operate in a network to produce cascading effects that contribute to the organization, development, and function of the immune system [26]. Multiplex assays allowed us to explore the immune status of breast milk in a single measurement. Moreover, multiplexed assays for immune compounds have not been used to investigate human milk from Asian women in a comparative study comprising various geographic, racial, socioeconomic, and environmental groups [27]. Thus, we added information about the cytokine profile in mature milk from Japanese women as an Asian population living in urban areas. The levels of immunologically active molecules are generally lower in mature milk than in colostrum [28]. However, the composition of mature milk may have a profound immunological effect in infants because high and long-term consumption of milk results in the cumulative exposure of infants to immune components [29]. Regarding the analytical aspect, compared with colostrum, mature milk was more suitable for observing the effects of probiotic supplementation on mothers regarding to smaller lactational changes in immune components.

Collectively, our study suggests that supplementation with a probiotic mixture during lactation, but not during pregnancy, modified the cytokine profile in mature milk. As dietary supplements are already popular among lactating women in Asian countries, this study also provides a simple regimen for implementing probiotic supplementation [30], one that does not require administration of probiotics to the infants.

A comparative study has shown milk cytokine levels in 10 regions across Africa, Europe, North America, and South America, suggesting the immunological composition of milk differs from one mother to another and likely reflects a mother’s ethnicities, living in geography, diet habits, socioeconomic status, and antigen exposure. [27]. The frequency of detection for 15 cytokines (IL-1β, -2, -4, -5, -6, -7, -8, 10, 12 (p70), -13, -17, TNF-α, MIP-1β, G-CSF, and GM-CSF) has been measured using the same analytical technique used in our study; thus, these results could be comparable to our study results. The total positive rates for IL-1β, TNF-α, IL-13, and IL-7 were lower than those reported in the literature (24% for IL-1β, 30% for TNF-α, 45% for IL-13, and 73% for IL-7), whereas those for IL-12 (p70) and GM-CSF were higher (75% for IL-12 (p70), 50% for GM-CSF) (Table 1) [27]. The levels of milk cytokines in Japanese women are also stated in another study using the same multiplex assay [31]. The level of IL-1β in this study was also lower than that reported in other countries in the comparative study. Taken together, low IL-1β appeared to be a characteristic feature of the milk cytokine profile among Japanese women. The detection rates of TNF-α, IL-13, IL-7, IL-12 (p70), and GM-CSF were different between populations, including in our results, suggesting these cytokines vary with population. In addition, the positive rate of IL-1RA and IL-7 were found to decrease with increasing periods of lactation in this study. Previous studies have shown a decrease in breast milk protein and cytokine expression with increasing lactation, especially in cytokine expression, which decreased rapidly until two months postpartum and then changed moderately [32]. A decrease in the positive rates of these cytokines seemed to reflect a decrease from one to two months and had a natural trend during lactation (Table 1).

Further, among cytokines with positive rates greater than 50%, the concentrations of 11 quantifiable cytokines were within the range for IL-8 and MIP-1β in each region in the comparative study (Table 2) [27]. However, the concentrations of IL-10 and GM-CSF were high in this study; in particular, IL-10 concentration was remarkably high compared with that reported in the literature [27,31]. Furthermore, probiotic supplementation had a time-dependent interaction effect on IL-10 concentration in breast milk. IL-10 plays a key role as a regulatory molecule in both innate and adaptive immune responses, contributing to developing immune tolerance and suppressing human tissue inflammation. Our study suggests that supplementation with the probiotic mixture can affect milk’s IL-10 levels. We have reported that TGF-β in breast milk may also have been affected by probiotic supplementation [13]. TGF-β and IL-10 are representative inhibitory cytokines produced by regulatory T cells [33], and changes in their levels in breast milk may reflect the maternal immune response to probiotic supplementation. Both LC5 and BG7 participate in immunomodulatory effects, including cytokine production in bacterially stimulated cells [20,22]. Supplemented probiotics appeared to interact locally in the gastrointestinal tract, whereas few bacterial cells could incorporate into the blood stream and then affect cells throughout body, including mammary gland epithelial cells. As milk cytokines are secreted from immune cells like monocytes and macrophages in milk as well as in mammary glands, the supplemented probiotic strains would affect both cytokine production through the local and systematic pathways involved in the production and secretion of human milk and the immune cells found in human milk.

In a prospective cohort study with probiotics, milk IL-10 concentration was high at three months postpartum in mothers who received the probiotics mixture (*L. rhamnosus, B. breve*, and *P. freudenreichii*) from gestational week 36 to delivery [17]. In contrast, no significant effect was observed in milk IL-10 concentration at three and six months postpartum in mothers who received single probiotics (*L. rhamnosus* or *B. lactis*) from two to five weeks before delivery [11]. Accordingly, supplementation with multiple probiotic strains appears to affect IL-10 concentration in breast milk rather than supplementation with a single strain. In addition, the result of repeated ANOVA shows that milk IL-10 were interactively affected by the time and intervention. The network related to IL-10 were differently visualized in time and intervention, implying that the mechanisms underlying the interactive change in the milk IL-10 were influenced by multiple signals among cytokines.

This study was limited because of its nature as an open-label study. This pilot study could not exclude the impact of potential placebo effects on cytokine and chemokine levels in breast milk, as the participants voluntarily selected the two groups. Given the potential differences in environmental and socioeconomic factors affecting the selection between these two groups, the immunological background associated with these factors may influence the levels of immune components in breast milk [27,34,35]. The cytokine levels in breast milk also vary according to maternal environmental factors such as ethnic factors, geographic location, dietary patterns, socioeconomic status, and psychosocial conditions. Because of voluntary selection, the unequal number of participants in both groups might have statistically influenced the outcome of this study. In particular, mothers who chose the probiotic group were more concerned about their health, and their diets and lifestyles may have influenced the results. *n*-3 PUFA supplements including fish oil are becoming increasingly popular in pregnant and lactating women in Japan and other East Asian countries [30], and their usage may affect immune components in breast milk [36]. However, detailed information on diet was not obtained in this study. In addition, food frequency questionnaires and dietary recall are available options to monitor the dietary patterns in lactating women. These limitations may affect the validity of the observed associations and should be considered when interpreting the study results. A randomized, placebo-controlled trial is thus needed to determine the overall impact of probiotic supplementation on breast milk cytokine concentrations in lactating mothers.

In conclusion, our results suggest that short-term probiotic intake in lactational mothers improves cytokine levels in breast milk while potential placebo effects still exist. Although we did not evaluate immunological outcomes in infants, milk IL-10 has immunomodulatory effects on the infant’s intestinal tract [8]. Therefore, IL-10 can assess the effects of probiotic supplementation on milk cytokine profiles.

## Figures and Tables

**Table 1 nutrients-13-02285-t001:** Composition of probiotic tablets.

Nutrients	Probiotic Tablets	Origin	Function and References
Energy (kcal)	1.2		
Protein (g)	0–0.1		
Fat (g)	0–0.1		
Carbohydrates (g)	0.7		
Sodium (mg)	0–2		
*Lactobacillus casei* LC5 (CFU)	5 × 10^9^	Cheese	Anti-inflammatory activity in lipopolysaccharide-activated RAW 264.7 cells [20].
*Bifidobacterium longum* BG7 (CFU)	5 × 10^9^	Infant feces	Suppresses *Helicobacter pylori*-induced interreukin-8 production in human gastric cell lines [22].
*Bacillus coagulans* SANK70258 (CFU)	2 × 10^8^	Green malt	Increase in persons whose defecation frequency [23].

Composition of probiotic tablets were adopted from Table 1 in Takahashi T, et al. Front Nutr. 2019 [13]. Values are shown for the daily dose of the probiotic tablets (three tablets of 750 mg per day).

**Table 2 nutrients-13-02285-t002:** Baseline characteristics of the study population.

	Total Population	Probiotic Group	Control Group	*p* Value
	(*n* = 60)	(*n* = 41)	(*n* = 19)	(Probiotic vs. Control)
Baseline characteristics				
Age (y)	33 (30–36)	33 (27–39)	33 (23–43)	0.981
Height (cm)	160 (155–164)	159 (150–169)	161 (154–168)	0.105
Weight (kg)	53 (49–57)	54 (45–63)	53 (46–61)	0.633
BMI (kg/m^2^)	21 (19–23)	21 (19–23)	20 (18–22)	0.137
Gestational age (w)	39 (38–40)	39 (37–41)	39 (37–41)	0.775
Infant birth weight (g)	3078 (2923–3354)	3110 (2642–3578)	3083 (2802–3274)	0.415
Days postpartum (d)	41 (37–46)	44 (34–48)	42 (32–52)	0.583
Clinical history of allergies				
Any	60/60 (100%)	41/41 (100%)	19/19 (100%)	1.000
Asthma	6/60 (10%)	3/41 (7%)	3/19 (16%)	0.370
Atopic dermatitis	22/60 (37%)	16/41 (39%)	6/19 (32%)	0.578
Allergic rhinitis	47/60 (78%)	34/41 (83%)	13/19 (68%)	0.578

Baseline characteristics of study population were adopted from Table 2 in Takahashi T, et al. Front Nutr. 2019 [13]. Values are shown as median (interquartile range) except for clinical history of allergies, which is shown as proportion (%). BMI, body mass index.

**Table 3 nutrients-13-02285-t003:** Effects of probiotic supplementation on the positive rates of cytokines.

Variable	Total Positive Rates (*n* = 180)	Probiotic Group (*n* = 41)			Control Group (*n* = 19)			Supplement	Time
[%]		1 Mo	2 Mo	3 Mo	1 Mo	2 Mo	3 Mo	Effect	Effect
Proinflammatory cytokines									
IL-1β	5.6	4.9	0	7.3	10.5	5.3	10.5	0.210	0.689
IL-6	30.6	41.5	17.1	31.7	42.1	31.6	21.1	0.838	0.115
IL-17	1.7	0	0	2.4	5.3	0	5.3	0.227	0.484
TNF-α	12.2	9.8	14.6	19.5	15.8	0	5.3	0.155	0.576
Anti-inflammatory cytokines									
IL-1RA	63.9	82.9	58.5	58.5	84.2	57.9	31.6	0.236	0.0002 *
IL-10	100	100	100	100	100	100	100	–	–
Th1-related cytokines									
IL-12 (p70)	100	100	100	100	100	100	100	–	–
Th2-related cytokines									
IL-4	1.7	0	0	2.4	5.3	0	5.3	0.227	0.484
IL-5	0	0	0	0	0	0	0	–	–
IL-13	7.8	0	7.3	7.3	5.3	21.1	15.8	0.039	0.093
Th9-related cytokines									
IL-9	100	100	100	100	100	100	100	–	–
Chemokines									
IL-8	99.4	100	100	100	94.7	100	100	0.996	0.995
Eotaxin	13.9	19.5	9.8	9.8	15.8	15.8	15.8	0.615	0.293
IP-10	93.9	95.1	92.7	90.2	94.7	94.7	100	0.332	0.703
MIP-1α	11.1	9.8	2.4	24.4	10.5	5.3	10.5	0.495	0.087
MIP-1β	97.2	100	95.1	92.7	100	100	100	0.997	0.124
RANTES	30.6	43.9	19.5	29.3	31.6	31.6	26.3	0.884	0.167
Growth factors									
IL-2	0	0	0	0	0	0	0	–	–
IL-7	60.6	70.7	58.5	48.8	68.4	68.4	52.6	0.622	0.026 *
FGF-basic	2.8	0	2.4	4.9	5.3	0	5.3	0.685	0.282
G-CSF	29.4	31.7	26.8	31.7	31.6	31.6	21.1	0.783	0.689
GM-CSF	54.4	53.7	53.7	51.2	57.9	52.6	63.2	0.527	1.000
PDGF-BB	91.1	95.1	85.4	85.4	100	94.7	94.7	0.100	0.113
VEGF	100	100	100	100	100	100	100	–	–

Positive rates of cytokines were analyzed using logistic regression. * *p <* 0.05. mo: month postpartum, IL: Interleukin: IL, TNF: tumor necrosis factor, RA: receptor antagonist, IP: interferon-gamma induced protein, MIP: macrophage inflammatory protein, RANTES: regulated on activation, normal T cell expressed and secreted, FGF: fibroblast growth factor, G-CSF: granulocyte colony-stimulating factor, GM-CSF: granulocyte-macrophage colony-stimulating factor, PDGF: platelet-derived growth factor, VEGF: vascular endothelial growth factor.

**Table 4 nutrients-13-02285-t004:** Effects of probiotic supplementation on levels of cytokines.

Variable [pg/mL, Mean 95% CI]	Probiotic Group (*n* = 41)			Control Group (*n* = 19)			Supplement × Time	Supplement	Time
	1 Mo	2 Mo	3 Mo	1 Mo	2 Mo	3 Mo	Interaction	Effect	Effect
Anti-inflammatory cytokines									
IL-1RA	366.33	169.95	296.49	405.15	128.742	339.51	0.854	0.916	0.132
	(188.76–543.89)	(108.28–231.62)	(−12.46–605.431)	(144.31–665.99)	(38.15–219.33)	(−114.32–793.34)			
IL-10	445.68	471.19	460.71	371.49	394.078	444.81	0.015 *	0.003 *	0.001 *
	(423.47–467.89)	(445.05–497.33)	(433.56–487.86)	(338.87–404.11)	(355.68–432.48)	(404.93–484.69)			
Th1-related cytokines									
IL-12 (p70)	411.91	438.36	425.15	371.43	379.89	381.57	0.467	0.003 *	0.094
	(391.84–431.97)	(419.52–457.21)	(403.91–446.38)	(341.96–400.91)	(352.21–407.58)	(350.38–412.76)			
Th9-related cytokines									
IL-9	37.32	29.58	40.07	37.99	30.42	38.38	0.919	0.992	0.070
	(29.02–45.61)	(25.85–33.31)	(28.13–52.00)	(25.80–50.17)	(24.95–35.91)	(20.85–55.91)			
Chemokines									
IL-8	85.59	65.13	152.50	132.25	110.53	772.53	0.214	0.172	0.116
	(38.57–132.59)	(29.96–100.30)	(−381.71–686.72)	(63.20–201.30)	(58.86–162.20)	(−12.22–1557.27)			
IP-10	2015.16	1206.67	1214.43	1936.73	1674.03	1157.72	0.667	0.878	0.152
	(837.40–3192.93)	(324.12–2089.22)	(451.34–1977.52)	(195.95–3677.52)	(369.52–2978.53)	(27.32–2288.12)			
MIP-1β	60.79	33.64	79.74	80.76	47.58	185.64	0.399	0.276	0.116
	(31.50–90.08)	(19.85–47.43)	(−41.59–201.06)	(37.73–123.78)	(27.32–67.84)	(7.43–363.86)			
Growth factors									
IL-7	25.01	20.67	21.47	23.34	31.75	23.16	0.073	0.438	0.390
	(19.11–30.91)	(14.51–26.83)	(14.17–28.77)	(14.67–32.01)	(22.70–40.80)	(12.44–33.88)			
GM-CSF	73.82	76.60	81.25	78.16	72.93	69.02	0.327	0.790	0.948
	(57.16–90.48)	(59.16–94.05)	(62.13–100.36)	(53.69–102.63)	(47.31–98.56)	(40.94–97.10)			
PDGF-BB	6.33	7.49	7.61	6.93	10.31	9.17	0.697	0.334	0.213
	(4.95–7.72)	(4.45–10.53)	(4.60–10.62)	(4.89–8.96)	(5.84–14.78)	(4.76–13.59)			
VEGF	8216.34	8807.23	8661.96	7226.86	7712.81	7886.36	0.695	0.041 *	0.011 *
	(7674.87–8757.81)	(8203.69–9410.76)	(8079.43–9244.48)	(6431.45–8022.26)	(6826.23–8599.39)	(7030.65–8742.08)			

The cytokines in the table are those with a total positive rate of 50% or higher in Table 1. * *p <* 0.05. mo: month postpartum, IL: Interleukin: IL, TNF: tumor necrosis factor, RA: receptor antagonist, IP: interferon-gamma induced protein, MIP: macrophage inflammatory protein, RANTES: regulated on activation, normal T cell expressed and secreted, FGF: fibroblast growth factor, G-CSF: granulocyte colony-stimulating factor, GM-CSF: granulocyte-macrophage colony-stimulating factor, PDGF: platelet-derived growth factor, VEGF: vascular endothelial growth factor.

## Data Availability

The datasets for this manuscript are not publicly available because they contain personal information. Requests to access the datasets should be directed to Tomoki Takahashi (tomoki-takahashi@beanstalksnow.co.jp).

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
