# Peer review of "Probiotic Supplementation and Human Milk Cytokine Profiles in Japanese Women: A Retrospective Study from an Open-Label Pilot Study"

_nutrients, 2021, doi:10.3390/nu13072285_

Round 1

Reviewer 1 Report

Dear Authors,

The study is important and interesting, however although it raises various concerns.

1) There is a need for a summary table including of the properties of the cytokines analysed in this work and their impact on child health.

2) As the study has a pilot character, its aims and hypotheses were inadequately formulated. Among others the aim of such a study is to calculate the sample size necessary for a full analysis, assessment of drop-outs, compliance, etc. The authors have already published pilot study on a similar topic. Without a good rationale for conducting further pilot studies analyzing the same subject, there is no scientific justification to do it. It looks like “salami slicing”.

3) The way of the allocation to groups should be described in methodology

4) Although "The detailed information regarding the probiotic tablets is presented in our previous study" it would be good to provide it in this manuscript. Especially important is how the CFU of the different strains in tablet were calculated.

5) It is worth preparing a table summarising the results; you have to consider to calculate the correlation between the obtained results and performing multivariate and network analyses.

6) Please discuss why cytokine analysis was not performed in mothers' blood; was active allergy excluded; was medication affected the immune system excluded?

7) Conclusions should be more general and take into account the pilot nature of the study.

Author Response

1) There is a need for a summary table including of the properties of the cytokines analysed in this work and their impact on child health.

We added a brief summary table adopted from a series of literature in Lines 39–49 in pages 1–2 in Introduction and Supplemental Table S1. Since milk cytokines and infant development has been extensively researched but present findings varies from the molecules, I would like you to understand the table as a kind of narrative review.

2) As the study has a pilot character, its aims and hypotheses were inadequately formulated. Among others the aim of such a study is to calculate the sample size necessary for a full analysis, assessment of drop-outs, compliance, etc. The authors have already published pilot study on a similar topic. Without a good rationale for conducting further pilot studies analyzing the same subject, there is no scientific justification to do it. It looks like “salami slicing”.

We appreciate your constructive criticism. To avoid from salami slicing-like appearance, we revised the rationales of this study as distinctive research from the previous study in Lines 39-49 in pages 1-2 in Introduction.

3) The way of the allocation to groups should be described in methodology

We added a brief methodology for the allocation in Lines 82-84 in page 2 in Material and Methods as follows:

To lessen the burden and dropout in the participants, the participants self-selected to join either the probiotic group (n = 41) or the no supplementation control group (n = 19).

4) Although "The detailed information regarding the probiotic tablets is presented in our previous study" it would be good to provide it in this manuscript. Especially important is how the CFU of the different strains in tablet were calculated.

We have prepared composition of probiotic tablets as Table.1 and added information regarding the reason to select the strains in the revised manuscript in lines 249-266, p.24 in Discussion as follows:

 " Three different combinations of probiotics were selected for this study. A mixture containing L. casei LC5, was isolated from cheese [51],  reduced NO production from the murine RAW264.7 cell line with bacterial lipopoly-saccharide, indicating that LC5 involves anti-inflammatory activity [17]. B. longum BG7, was isolated from infant fe-ces [52], suppresses H. pylori-induced IL-8 production in human gastric cell lines [18], implying that BG7 can affect cytokine production induced by bacterial infection. B. coagulans SANK70258, was isolated from green molt , improves the intestinal envi-ronment in healthy adults, as evidenced by an increase in bifidobacteria and a decrease in Clostridium perfringens [19]. Given these informations, these probiotic strains may exert beneficial effects on health by modifying the immune system. Furthermore, a previous report showed that a mixture of different probiotic genera, Bifidobacterium and Lactobacillus, might be more effective in preventing allergic disease of infants than a single genus; in this report, the subjects were supplemented with ≧109 CFU in many trials (14). In addition, for the purpose of increasing adherence by providing a percep-tible benefit to the mother herself, we included 108 CFU tablets of B. coagulans SANK70258 as the amount of evidence for improving bowel movements [19]. Thus, we studied a combination of these three species with 5 × 109 CFU of Lactobacillus casei LC5, 5 × 109 CFU of Bifidobacterium longum BG7, and 2 × 108 CFU of Bacillus coagulans SANK70258."

5) It is worth preparing a table summarising the results; you have to consider to calculate the correlation between the obtained results and performing multivariate and network analyses.

We prepared network analysis and add in Supplementary Figure S2.

We added the result in Lines 223-224 in page 19 in Results as follows:

“ The topology of networks of cytokine between probiotic and control were visually different (Supplementary Figure S2).”

6) Please discuss why cytokine analysis was not performed in mothers' blood; was active allergy excluded; was medication affected the immune system excluded?

To lessen the mother's burden of participating the study and ethical consideration for invasive sampling, regular visits to the hospital and blood sampling were not conducted. In addition, previous study was reported to not correlate with TGF-β, is the main target of analysis in our previous study, in maternal blood and milk (Hawkes JS, et al. Cytokine. 2002.).

Active allergies were excluded from the current study because the purpose of the study was to examine the preventive effect of probiotic intervention on a relatively healthy population, rather than active treatment for allergic patients.

In addition, medication use was not excluded because of ethical considerations.

7) Conclusions should be more general and take into account the pilot nature of the study.

We revised conclusion in accordance with the reviewer’s suggestion in Lines 26-27 in page 1 in Abstract and in Lines 354-355 in page 26 in Discussion as follows:

“A short-term supplementation of probiotics affect human milk cytokine levels in lactating women while possible placebo effect still exist. “

“Overall, our results suggest that short-term probiotic intake in lactational mothers improves cytokine levels in breast milk while potential placebo effects still exist.”

Reviewer 2 Report

The study was designed to determine the impact of probiotic supplementation during early lactation on milk cytokine levels. It is a simplistic study with a few imitations but it has sound analyses.

A few suggestions in terms of being more transparent regarding methodology:

1) Were the samples analyzed in duplicates? If not, were inter and intra-CV rates calculated for the 27-plex assay?

2) Were the 3 bacteria strains used in the supplementation study already found in human milk? the genera Bifidobacterium and Lactobacillus have been reported. But what about the species used? Perhaps it would be helpful to add the reasoning as to why these strains were selected.

3) Is there information on the milk oligosaccharide secretor/non-secretor status of the mothers? Would it be possible that the presence of specific oligosaccharides in milk has an impact on milk cytokine changes?

Author Response

The study was designed to determine the impact of probiotic supplementation during early lactation on milk cytokine levels. It is a simplistic study with a few imitations but it has sound analyses.

A few suggestions in terms of being more transparent regarding methodology:

Thank you for your kind consideration on our manuscript. Point-to-point replies were enclosed below.

1) Were the samples analyzed in duplicates? If not, were inter and intra-CV rates calculated for the 27-plex assay?

Yes, we measured each samples in duplicate.

2) Were the 3 bacteria strains used in the supplementation study already found in human milk? the genera Bifidobacterium and Lactobacillus have been reported. But what about the species used? Perhaps it would be helpful to add the reasoning as to why these strains were selected.

As far as I know, L. casei and B. longum are present in breast milk(Soto A, et al. J Pediatr Gastroenterol Nutr. 2014.), but I am not sure if B.coagulans is present.

We have prepared composition of probiotic tablets as Table.1 and added information regarding the reason to select the strains in the revised manuscript in lines 249-266, p.24 in Discussion as follows:

 " Three different combinations of probiotics were selected for this study. A mixture containing L. casei LC5, was isolated from cheese [51], reduced NO production from the murine RAW264.7 cell line with bacterial lipopoly-saccharide, indicating that LC5 involves anti-inflammatory activity [17]. B. longum BG7, was isolated from infant fe-ces [52], suppresses H. pylori-induced IL-8 production in human gastric cell lines [18], implying that BG7 can affect cytokine production induced by bacterial infection. B. coagulans SANK70258, was isolated from green molt , improves the intestinal envi-ronment in healthy adults, as evidenced by an increase in bifidobacteria and a decrease in Clostridium perfringens [19]. Given these informations, these probiotic strains may exert beneficial effects on health by modifying the immune system. Furthermore, a previous report showed that a mixture of different probiotic genera, Bifidobacterium and Lactobacillus, might be more effective in preventing allergic disease of infants than a single genus; in this report, the subjects were supplemented with ≧109 CFU in many trials (14). In addition, for the purpose of increasing adherence by providing a percep-tible benefit to the mother herself, we included 108 CFU tablets of B. coagulans SANK70258 as the amount of evidence for improving bowel movements [19]. Thus, we studied a combination of these three species with 5 × 109 CFU of Lactobacillus casei LC5, 5 × 109 CFU of Bifidobacterium longum BG7, and 2 × 108 CFU of Bacillus coagulans SANK70258."

3) Is there information on the milk oligosaccharide secretor/non-secretor status of the mothers? Would it be possible that the presence of specific oligosaccharides in milk has an impact on milk cytokine changes?

No, we did not measure it in our study. In other small sub-cohort study, we confirmed approximately 90% were secretor in the lactating women in Japan (n=40). This cross-sectional data were presented electronically in the conference in Japan Society for Bioscience, Biotechnology, and Agrochemistry 2021. Accordingly, we estimate that secretor mother appears to be slightly higher in Japan than western population.

The ability to consume HMOs is different for each member of human milk flora that they release a variety of cytokines into the milk. Therefore, breast milk HMO composition may influence cytokine composition in breast milk via changes in breast milk flora composition. We added the discussion in Line 320-328 in page 25 in the Discussion as follows :

“Both LC5 and BG7 involve in the imunomodulating effects including cytokine produc-tion in bacterially stimulated cells [17,18]. Supplemented probiotics appeared to inter-act locally in the gastrointestinal tract, while a limited bacterial cells could be incor-porated in blood stream and then affect throughout body including mammary gland epithelial cells. Since milk cytokines are secreted from immune cells like monocytes and macrophages in milk as well as mammary glands, the supplemented probiotic strains would affect cytokine production through the local and systematic path-ways in the production and secretion of human milk and the immune cells found in human milk.”

Reviewer 3 Report

The manuscript “Probiotic supplementation and human milk cytokine profiles in Japanese women: a retrospective study from an open-label pilot study” is an interesting open-label study assessing probiotic supplementation to lactating mothers. The aim of the study was to check the changes in milk cytokine profile after the intervention with new combination of probiotic strains (Lactobacillus casei LC5, Bifidobacterium longum BG7, Bacillus coagulans SANK70258). 

The authors confirmed significant difference (increase) in IL-10 and IL-12 human milk concentration between intervention and control group.

The results are interesting and important, but the most visible flaw of the study is no information do the mothers in the intervention group had really the same diet and/or diet supplements as the control group. The study was open-label and the decision to participate in active arm (probiotics) was made by the mothers. This way there is potential bias, as these “active” mothers probably are more focused on healthy diet and supplements as for example omega-3 acids. This flaw was partially stated by the Authors, but I would add the comment about possible impact of more or less healthy diet and the information do (and if yes – how?) the mothers’ diet was monitored or checked.

The Authors underlined the impact of “ethnic factors, geographic location, dietary patterns, socioeconomic status, and psychosocial conditions” on cytokine levels in breast milk, but I would add some citations and data concerning the mothers’ diet as the factor influencing cytokines levels in breast milk.

Very interesting is the result that supplementation with multiple probiotic strains appears to affect IL-10 in breast milk more than supplementation with a single strain. I would discuss more widely what is the suspected mechanism of this impact of probiotics mixture on cytokines milk concentration. May we have more information about probiotic strains used in the study – were they investigated in vitro before? Do the Authors know/mean the IL-10 and IL-12 are simply produced by the bacteria strains supplemented to the mothers or do they think about other mechanisms? Which mechanisms are probable link between probiotics supplementation and cytokines changes in human milk? It would be very interesting to discuss it.

In summary, the paper is of good quality and worth publication after these minor changes.

Author Response

The manuscript “Probiotic supplementation and human milk cytokine profiles in Japanese women: a retrospective study from an open-label pilot study” is an interesting open-label study assessing probiotic supplementation to lactating mothers. The aim of the study was to check the changes in milk cytokine profile after the intervention with new combination of probiotic strains (Lactobacillus casei LC5, Bifidobacterium longum BG7, Bacillus coagulans SANK70258).

The authors confirmed significant difference (increase) in IL-10 and IL-12 human milk concentration between intervention and control group.

The results are interesting and important, but the most visible flaw of the study is no information do the mothers in the intervention group had really the same diet and/or diet supplements as the control group. The study was open-label and the decision to participate in active arm (probiotics) was made by the mothers. This way there is potential bias, as these “active” mothers probably are more focused on healthy diet and supplements as for example omega-3 acids. This flaw was partially stated by the Authors, but I would add the comment about possible impact of more or less healthy diet and the information do (and if yes – how?) the mothers’ diet was monitored or checked.

The Authors underlined the impact of “ethnic factors, geographic location, dietary patterns, socioeconomic status, and psychosocial conditions” on cytokine levels in breast milk, but I would add some citations and data concerning the mothers’ diet as the factor influencing cytokines levels in breast milk.

Thank you for reviewing our manuscript and informative suggestions. As you pointed out, the mothers who chose the probiotic group had a high level of health concern, and their diet and lifestyle may reflect this as a potential bias. Therefore, we have added information dietary influence on breast milk cytokine in the revised the manuscript in lines 345-350, p.25 as follows: " In particular, mothers who chose the probiotic group were more concerned about their health, and their diets and lifestyles may have influenced the results. n-3 PUFA sup-plements including fish oil are becoming increasingly popular in pregnant and lactat-ing women in Japan and other East Asian countries [57] and it may affect immune components in breast milk [62]. However, detailed information on diet was not ob-tained in this study. "

Very interesting is the result that supplementation with multiple probiotic strains appears to affect IL-10 in breast milk more than supplementation with a single strain. I would discuss more widely what is the suspected mechanism of this impact of probiotics mixture on cytokines milk concentration. May we have more information about probiotic strains used in the study – were they investigated in vitro before? Do the Authors know/mean the IL-10 and IL-12 are simply produced by the bacteria strains supplemented to the mothers or do they think about other mechanisms? Which mechanisms are probable link between probiotics supplementation and cytokines changes in human milk? It would be very interesting to discuss it.

In summary, the paper is of good quality and worth publication after these minor changes.

We added the discussion considering the reviewer’s suggestions in Lines 320–328 in page 25 in Discussion as follows:

“Both LC5 and BG7 involve in the imunomodulating effects including cytokine produc-tion in bacterially stimulated cells [17,18]. Supplemented probiotics appeared to inter-act locally in the gastrointestinal tract, while a limited bacterial cells could be incor-porated in blood stream and then affect throughout body including mammary gland epithelial cells. Since milk cytokines are secreted from immune cells like monocytes and macrophages in milk as well as mammary glands, the supplemented probiotic strains would affect cytokine production through the local and systematic path-ways in the production and secretion of human milk and the immune cells found in human milk.”

次のように、ディスカッションの 25 ページの 320 行目から 328 行目にレビューアーの提案を考慮したディスカッションを追加しました。

「LC5 と BG7 は両方とも、細菌によって刺激された細胞におけるサイトカイン産生を含む免疫調節効果に関与しています [17,18]。補給されたプロバイオティクスは、消化管で局所的に相互作用するように見えましたが、限られた細菌細胞は血流に取り込まれ、乳腺上皮細胞を含む体全体に影響を与える可能性があります.牛乳のサイトカインは、母乳中の単球やマクロファージ、乳腺などの免疫細胞から分泌されるため、補充されたプロバイオティクス株は、人乳の産生および分泌における局所的および系統的な経路を通じてサイトカイン産生に影響を及ぼします。人乳」。

Reviewer 4 Report

The manuscript by Tomoki Takahashi et al. is a report, which compares the interleukins content  in human milk from 1st, 2nd and 3rd month of lactation. The manuscript present an important aspect of immunological status of human milk, but in my opinion, the result section wasn’t prepare property, while introduction and discussion sections do not present the discussed issues exhaustively. Below, I present the most important aspects of the work that, in my opinion, require extensive improvement.

  1. The Introduction section should describe the biological importance of molecules analyzed in the manuscript .  

In the manuscript should contain the information, that: The immune factors present in human milk such as cytokines, chemokines, and growth factors contribute to differentiation of IgA-producing cells, playing a pivotal role in the maturation of the infant GI-associated immune system and in protecting the newborn against infectious diseases. Moreover, the authors should describe mechanism working of pro-inflammatory and anti-inflammatory cytokines and growth factors.

Generally, in my opinion the introduction doesn't provide sufficient background and include all relevant references.

  1. The authors should explain why probiotic supplementation is important for pregnant and lactating women. Moreover, the authors, should argue, why the decided on supplementation of Bacillus coagulans, Bifidobacterium longum and Lactobacillus casei
  2. The information about pre-treatment of human milk samples for analysis (i.e dilution of milk sample, used buffers) and are extremely poor. The authors mentioned only that “the assay was conducted according to the manufacturer’s instructions”. In my opinion, it is insufficient.
  3. The results section was not properly described and the results are not clearly presented.
  • Background characteristics of mothers and infants are extremely poor. I understand that in detail it was reported previously, but general information about study groups such as, maternal age, BMI, and health problems  should be described by authors.
  • The authors should consider the preparation of the heatmap or box-plot diagrams, representing the median concentration of different immune factors (cytokines, growth factor, and chemokines) in human milk samples from 1st, 2nd, 3rd month of lactation.
  • Table 1: The results are difficult to interpret. I do not understand why the authors do not show the concentration of immune factors in analyzed samples of milk.

Thank you for the opportunity to review the manuscript titled "Probiotic supplementation and human milk cytokine profiles in Japanese women: a retrospective study from an open-label pilot study". I hope that the proposed changes will help to improve the manuscript.

Author Response

The manuscript by Tomoki Takahashi et al. is a report, which compares the interleukins content  in human milk from 1st, 2nd and 3rd month of lactation. The manuscript present an important aspect of immunological status of human milk, but in my opinion, the result section wasn’t prepare property, while introduction and discussion sections do not present the discussed issues exhaustively. Below, I present the most important aspects of the work that, in my opinion, require extensive improvement.

The Introduction section should describe the biological importance of molecules analyzed in the manuscript . 

In the manuscript should contain the information, that: The immune factors present in human milk such as cytokines, chemokines, and growth factors contribute to differentiation of IgA-producing cells, playing a pivotal role in the maturation of the infant GI-associated immune system and in protecting the newborn against infectious diseases. Moreover, the authors should describe mechanism working of pro-inflammatory and anti-inflammatory cytokines and growth factors.

Generally, in my opinion the introduction doesn't provide sufficient background and include all relevant references.

We added a summary table and revised a brief introduction in Lines 39–49 in pages 1–2 in Introduction and Supplemental Table 1 in accordance with the reviewers’ suggestion.

The authors should explain why probiotic supplementation is important for pregnant and lactating women. Moreover, the authors, should argue, why the decided on supplementation of Bacillus coagulans, Bifidobacterium longum and Lactobacillus casei

We added the explanation for the importance to supply probiotics in pregnant and lactating women in Lines 39-49 in page 1-2 in Introduction, and described the details of supplementation in Table 1.

In addition, we have added information regarding the reason to select the strains in the revised manuscript in lines 249-266, p.24 in Discussion as follows:

 " Three different combinations of probiotics were selected for this study. A mixture containing L. casei LC5, was isolated from cheese [51],  reduced NO production from the murine RAW264.7 cell line with bacterial lipopoly-saccharide, indicating that LC5 involves anti-inflammatory activity [17]. B. longum BG7, was isolated from infant fe-ces [52], suppresses H. pylori-induced IL-8 production in human gastric cell lines [18], implying that BG7 can affect cytokine production induced by bacterial infection. B. coagulans SANK70258, was isolated from green molt , improves the intestinal envi-ronment in healthy adults, as evidenced by an increase in bifidobacteria and a decrease in Clostridium perfringens [19]. Given these informations, these probiotic strains may exert beneficial effects on health by modifying the immune system. Furthermore, a previous report showed that a mixture of different probiotic genera, Bifidobacterium and Lactobacillus, might be more effective in preventing allergic disease of infants than a single genus; in this report, the subjects were supplemented with ≧109 CFU in many trials (14). In addition, for the purpose of increasing adherence by providing a percep-tible benefit to the mother herself, we included 108 CFU tablets of B. coagulans SANK70258 as the amount of evidence for improving bowel movements [19]. Thus, we studied a combination of these three species with 5 × 109 CFU of Lactobacillus casei LC5, 5 × 109 CFU of Bifidobacterium longum BG7, and 2 × 108 CFU of Bacillus coagulans SANK70258. "

The information about pre-treatment of human milk samples for analysis (i.e dilution of milk sample, used buffers) and are extremely poor. The authors mentioned only that “the assay was conducted according to the manufacturer’s instructions”. In my opinion, it is insufficient.

We have added information regarding the pre-treatment of milk samples for analysis in the revised manuscript in 118-123, p.4 in Material and Methods as follows:

" All skimmed milk samples were diluted 4-fold with sample diluent (50 µL skimmed milk + 150 µL sample diluent) prior to use and pipetted into 96-well plates. The assay was conducted according to the manufacturer’ s instructions and all samples were an-alyzed in duplicates. The content of each microplates were read using a Bio-Plex 200 System plate reader (Bio-Rad Laboratories) and determined using Bio-Plex Manager software, version 5.0."

The results section was not properly described and the results are not clearly presented.

Background characteristics of mothers and infants are extremely poor. I understand that in detail it was reported previously, but general information about study groups such as, maternal age, BMI, and health problems  should be described by authors.

Thank you for the suggestion. We prepared baseline characteristics of study population as Table 2, as adopted from our previous report.

The authors should consider the preparation of the heatmap or box-plot diagrams, representing the median concentration of different immune factors (cytokines, growth factor, and chemokines) in human milk samples from 1st, 2nd, 3rd month of lactation.

According to the reviewer’s suggestion, we prepared box-plot diagrams as Supplementary Figure S1.

Table 1: The results are difficult to interpret. I do not understand why the authors do not show the concentration of immune factors in analyzed samples of milk.

As same to the previous comments and its response, we added the concentrations in supplementary materials. Please refer to Supplementary Table S2 & 3.

Thank you for the opportunity to review the manuscript titled "Probiotic supplementation and human milk cytokine profiles in Japanese women: a retrospective study from an open-label pilot study". I hope that the proposed changes will help to improve the manuscript.

Thank you for your suggestion on our manuscript. The reviewer’s comments help us to greatly improve the manuscript.

Round 2

Reviewer 1 Report

Dear Authors,

The manuscript was significantly improved, however few subject was not addressed by authors.

1) Due to pilot character of this study, its aims and hypotheses should be rewritten. Among others the aim of such a study is to calculate the sample size necessary for a full analysis, assessment of drop-outs, compliance, etc.

2) The network analyses should show type (positive, negative) and weight of ralation. It must be detailed described and discussed.

3) Conclusions should take into account the pilot nature of the study – see point 1.

Author Response

1) Due to pilot character of this study, its aims and hypotheses should be rewritten. Among others the aim of such a study is to calculate the sample size necessary for a full analysis, assessment of drop-outs, compliance, etc.

We revised the manuscript in accordance with the reviewer’s suggestion in Lines 29-30 in page 1 in Abstract, in Lines 66-79 in page 2 in Introduction as follows:

“Future placebo-controlled studies are needed to support these results, based on the estimated sample sizes in this study.”

“When evaluating the immunological effects of supplementation with a new probiotic strain or combination of strains, the pilot study is important for improvement of the quality and efficiency of the main study [13]. Our previous pilot study suggested that a mixture of new probiotic strains (Lactobacillus casei LC5, Bifidobacterium longum BG7, and Bacillus coagulans SANK70258) was tolerated in lactating mothers with respect to no dropout and 91.5% adherence and possibly affected TGF-β and IgA levels in milk [14]. Since previous studies have investigated only a limited range of cytokines [12,15-19]. Considering the complex network interaction among cytokines, investigating a greater number of cytokines may help explain the contradiction in results between these studies.

In this study, 27 cytokines were comprehensively measured in breast milk samples collected in a previous study to calculate the sample size necessary for a full analysis and determine the effect of the new probiotic strains, L. casei LC5, B. longum BG7, and B. coagulans SANK70258, on other breast milk cytokines.”

2) The network analyses should show type (positive, negative) and weight of ralation. It must be detailed described and discussed.

Unfortunately, we were difficult to develop the figures with colours for positivity and weight.

Alternatively, we added hierarchical heat maps in supplementary Figure S3 (page 11) to refer to the positivity, weight and grouping. 

We added the description in Lines 191-193 in page 9 as follows:

“The levels of 24 cytokines were converted to common logarithm values to visualize the distribution of the data. The network analysis and hierarchical heat maps derived from Pearson's correlation coefficient.”

3) Conclusions should take into account the pilot nature of the study – see point 1.

Please see our response for point 1.

Reviewer 4 Report

The proposed revision (Round 1) changes were made in manuscript. In my opinion, they significantly increased the value of the manuscript. Below, I present the list of topics that might be improved, however, I am requesting minor edits. 

  1. The introduction section is not inconsistent. Some information has already been provided in the manuscript text and needs to be organized and expanded. I recommend beginning with general information about human milk and the importance of breastfeeding. Later, the information about immunologically bioactive proteins and other molecules and finally cytokines, chemokines and growth factors.
  2. Line 41-42 and 42-43. Please modify the sentences. It is not clear what the authors mean. 
  3. Line 119: The time of plate incubation during the 1st step of experiment (with milk samples)  is missing. Moreover, the information about the detection system is missing. Please describe in detail an immuno-assay.
  4. Line 249-266: The major part of this paragraph should be transferred to the Method section.
  5. The names of the bacteria should be in italics; please correct throughout the manuscript.
  6. Line 298-299: Can the authors briefly comment on these results?
  7. Do the results have implications for milk biobanking or neonatal care for premature newborns?

Author Response

  1. The introduction section is not inconsistent. Some information has already been provided in the manuscript text and needs to be organized and expanded. I recommend beginning with general information about human milk and the importance of breastfeeding. Later, the information about immunologically bioactive proteins and other molecules and finally cytokines, chemokines and growth factors.

We added the general information about breastfeeding in accordance with the reviewer’s suggestion in Lines 35-37 in page 1 in beginning of Introduction as follows:

“Breastfeeding provides the ideal nutrition for infant growth and development. It can protects against gastrointestinal or respiratory tract infections[1] and reduce the risk for diarrhea morbidity or mortality in early infancy [2].”

  1. Line 41-42 and 42-43. Please modify the sentences. It is not clear what the authors mean.

We deleted the part corresponding to Lines 41-43 (Lines 46-48 in revised manuscript) in page 1as follows:

“All compounds and cells in milk play their nutritional roles in infants like a biological system [6]. For cellular compounds, immune cells and immune-active proteins appear to act interactively in human milk”

 I explained it at the beginning in Introduction.

  1. Line 119: The time of plate incubation during the 1st step of experiment (with milk samples) is missing. Moreover, the information about the detection system is missing. Please describe in detail an immuno-assay.

We added detailed instructions about the bio-plex assay, is similar to that of a sandwich ELISA, to Line 144-160 in page 5 as follows:

“Briefly, vortex the magnetic beads (1×) for 30 sec at medium speed and then were added 50 μL to each well of the assay plates. The plate was washed with 100 μL wash buffer per well. Vortex the diluted skimmed milk samples, cytokine standards, blank at medium speed for 5 sec and then were added the well in duplicates of the plate. The assay plate was covered with sealing tape and incubated on shaker at 850 ± 50 rpm for 30 min at room temperature. After incubating, the plate was washed three times with 100 µL wash buffer. Vortex the diluted detection antibodies (1×) at medium speed for 5 sec and then were added 25 µL to each well. The plate was covered and incubated on shaker at 850 ± 50 rpm for 30 min at room temperature. After detection antibody in-cubation, the plate was washed three times with 100 µL wash buffer. Vortex the dilut-ed Streptavidin-Phycoerythrin (SA-PE) (1×) at medium speed for 5 sec and then were added 50 µL to each well. After the SA-PE incubation, the plate was washed three times with 100 µL wash buffer. To resuspend beads for plate reading, add 125 µL assay buffer to each well. The plate was covered with sealing tape and shaked at room tem-perature at 850 ± 50 rpm for 30 sec. After shaking step, The assay plates were read using a Bio-Plex 200 System plate reader (Bio-Rad Labor-atories) and determined using the Bio-Plex Manager software, version 5.0.”

  1. Line 249-266: The major part of this paragraph should be transferred to the Method section.

The commented paragraph were transferred to Materials and Methods section in Lines 95-114 in page 2-3 in accordance with the reviewer’s suggestion.

  1. The names of the bacteria should be in italics; please correct throughout the manuscript.

We corrected the name of the bacteria in italics in revised manuscript.

  1. Line 298-299: Can the authors briefly comment on these results?

We added briefly comment in Lines 357-361 in page 27 in Discussion as follows:

“A comparative study has shown milk cytokine levels in 10 regions across Africa, Europe, North America, and South America, and suggested that the immuno-logical composition of milk differs from one mother to another and likely reflects a mother’s ethnicities, living in geography, diet habits, socioeconomic status, and antigen exposure. [58].”

  1. Do the results have implications for milk biobanking or neonatal care for premature newborns?

Thank you for your insightful comments. In my opinion, some NICU routinely uses probiotics in the infants under specific conditions such as high risk patients in preterm birth. However, mothers are not applicable for the supplementation of probiotics. When more evidences are developed, probiotics supplementation for mothers can be available to manage the quality of  donor milk regarding to bio active proteins like cytokines.